# Remote Sensing detectability of airborne Arctic dust

Norman T. O'Neill[1], Keyvan Ranjbar[2], Liviu Ivănescu[3], Yann Blanchard[4], Seyed Ali Sayedain[1], Yasmin AboEl-Fetouh[5]

[1]Centre d'Applications et de Recherches en Télédétection, Université de Sherbrooke, Sherbrooke, Canada
[2]Flight Research Laboratory, National Research Council Canada, Ottawa, Canada
[3]Metrology Research Centre, National Research Council Canada, Ottawa, Canada
[4]Centre pour l'Étude et la Simulation du Climat à l'Échelle Régionale, Département des sciences de la Terre et de l'atmosphère, Université du Québec à Montréal, Montréal, Canada
[5]Institute of Meteorology and Climate Research, Karlsruhe Institute of Technology, Karlsruhe, Germany

*Correspondence to:* N. T. O'Neill (Norman.T.ONeill@USherbrooke.ca)

**Abstract.** Remote sensing (RS) based estimates of Arctic dust are oftentimes overestimated due to a failure in separating out the dust contribution from that of spatially homogeneous clouds or low-altitude cloud-like plumes. A variety of illustrations are given with a particular emphasis on questionable claims of using brightness temperature differences (BTDs) as a signature indicator of Arctic dust transported from mid-latitude deserts or generated by local Arctic sources. While there is little dispute about the presence of both Asian and local dust across the Arctic, the direct RS detectability of airborne dust, as ascribed to satellite (MODIS and AVHRR) measurements of significantly negative brightness-temperature differences at 11 and 12 µm ($BTD_{11-12}$) has been misrepresented in certain cases. While it is difficult to account for all examples of strongly negative $BTD_{11-12}$ values in the Arctic, it is unlikely that airborne dust plays a significant role. One, much more likely contributor would be water clouds in the Arctic inversion layer.

The RS detectability of the impact of Arctic dust (notably due to Arctic dust from local sources) can, however, be of significance. Sustained dust deposition can substantially decrease (visible to shortwave IR) snow and ice reflectance albedo (pan-chromatic reflectance) and the signal measured by satellite sensors. Significantly negative $BTD_{11-12}$ values would however only represent a limited area near the drainage basin sources according to our event-level case studies. The enhanced INP (Ice Nucleating Particle) role of local Arctic dust can, for example, induce significant changes in the properties of low-level mixed-phase clouds (cloud optical depth changes $<\sim 1$) that can be readily detected by active and passive RS instruments. It is critical that the distinction between the RS detectability of airborne Arctic dust versus the RS detectability of the impacts of that dust be understood if we are to appropriately parameterize, for example, the radiative forcing influence of dust in this climate sensitive region.

## 1 Introduction

Vincent (2018) (VCT) reported on the use of MODIS and AVHRR thermal infrared (TIR) brightness temperature differences (BTDs) in the western Canadian Arctic (Beaufort Sea and Amundsen Gulf region) to detect the presence of "persistent low-

level dust clouds" and dust deposited on ice, snow and water. A later publication (Bowen & Vincent, 2021) (B&V) argued that negative $BTD_{11-12}$ ($BT_{11\ \mu m} - BT_{12\ \mu m}$) values were a unique signature of dust (without explicitly distinguishing between airborne and surface deposited dust) and that this measure could be directly used to estimate the relative spatial extent of dust

in the Arctic. Those two water bodies (along with other place names and geographic features that are discussed below) appear in the Figure 1 map of the grey-shaded Canadian Arctic Archipelago (CAA) and associated Arctic and sub-Arctic regions.

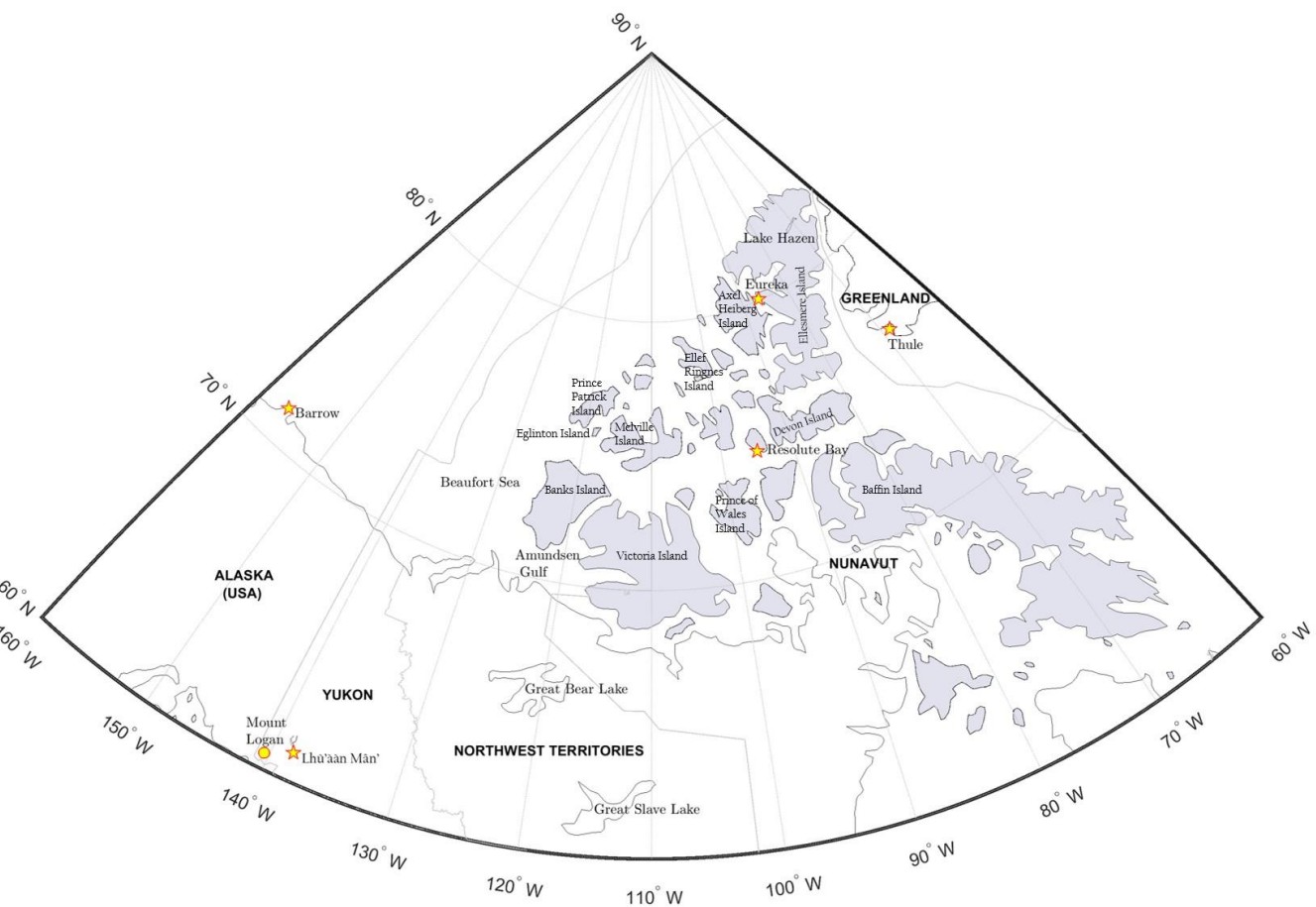

Figure 1 – Map of the (grey-shaded) islands of the Canadian Arctic Archipelago (CAA). The map also includes Arctic and subarctic research sites in Alaska, northern Canada and Greenland (indicated with golden stars) and geographic features that are discussed in the main text.


We acknowledge the robustness of negative $BTD_{11-12}$ values as a potential indicator of optically thick, airborne dust embedded in a normal-lapse-rate atmospheric layer or as a sign of accumulated dust deposition. We disagree with the affirmation that airborne dust clouds of Asian origin were commonly detected using passive, satellite-based remote sensing (RS). Springtime Asian dust, while representing a robust pan-Arctic seasonal event, yields, on average, very weak coarse mode (CM), roughly

submicron, aerosol optical depths (AOD) at 550 nm[1]. The multi-year, six-station, pan-Arctic, AERONET/AEROCAN monthly-binned (geometric means) climatology of AboEl-Fetouh et al. (2020) (AeF) indicate (their Figure 7) that the Resolute Bay CM AODs are largely of the greatest amplitude across the six stations: using that site as a reference, their CM AODs were $<\sim$ the Resolute Bay value ($0.016 \times 1.5^{\pm 1}$ during the Asian-dust dominated April, May springtime and $<\sim 0.0023 \times 1.2^{\pm 1}$ during the June, July, August (likely local dust) summertime. DODs (Dust Optical Depths) will be $\sim$ CM AODs in the absence

of any other significant CM source such as sea-salt or volcanic ash (or CM cloud in the absence of proper cloud screening). Springtime Asian dust aerosols at such small CM AOD (DOD) values are difficult if not impossible to detect using passive satellite-based RS techniques at any wavelength. The $BTD_{11-12}$ variation per unit change in DOD is $\sim$ -0.3 °K (see the discussion of Figure B1 for details). The corresponding change in $BTD_{11-12}$ for a springtime Asian dust DOD of 0.016 (the Resolute Bay maximum) would be an undetectable -0.005 °K (an amplitude that is significantly less than the nominal MODIS $BTD_{11-12}$

noise figure of 0.07 °K (again, see the discussion of Figure B1).

    While the monthly-averaged springtime Asian dust DODs are $<\sim 0.016$ there are springtime (mid-tropospheric) Asian dust events that do lead to more substantive DODs over the Arctic. In general these are limited to a few notable days in a given year, with individual DODs being $<\sim 0.4$: the roughly week-long dust intrusion over the Mount Logan (Yukon territory of Canada) in April of 2001 (DODs $<\sim 0.3$)[2], Stone et al. (2007) for the roughly week long, mid-tropospheric dust intrusions of

April 2002 over Barrow, Alaska (DODs $<\sim 0.4$), Zhao et al. (2022) for moderate DODs ($<\sim 0.1$) associated with single-day intrusions over Barrow in April of 2015, and Thulasiraman et el. (2002) for evidence that the sub-0.4 DODs of the April, 2001 event were arguably a broad west coast phenomenon stretching from (at least) low-Arctic Canada to the southern United States. DODs $\sim 0.4$ could incite a marginally detectable $BTD_{11-12}$ signal (-0.3 °K $\times 0.4 \sim$ -0.12 °K) and would be more easily identified in true-color and AOD imagery (at least over water).

VCT noted that a second dust source could have been locally generated dust storms (although dust from southern latitudes was claimed to be the major source). Indeed, Meinander et al. (2022) recently reviewed the importance of high latitude dust generated from local sources. However, even strong local dust plumes will likely not induce large DODs beyond the short temporal and spatial window associated with their detectable plume presence. Outside this window, the monthly averaged DOD upper limit reported by AeF make it very unlikely that DODs could be detected using passive, satellite-based RS in the

thermal infrared red (TIR).

    Empirical support for this affirmation is provided, for example, by Ranjbar et al.'s (2021) detection of a high-Arctic dust plume near its drainage-basin source: their MODIS $BTD_{11-12}$ values (amplitude $\sim 1.5$ °K) are $\sim$ the amplitude of the most extreme negative values ($\sim$ -1.0 °K) reported by VCT. However, Ranjbar's visible (532 nm) DODs were $\sim 0.5$ (a $BTD_{11-12}$ to DOD sensitivity of $dBTD_{11-12}/dDOD \sim 1.5 / 0.5 = 3$ °K per unit DOD). The AeF Resolute Bay summertime[3] CM AOD maximum

---

[1] Note that, unless otherwise stated, our AODs and DODs will always be referenced to a wavelength of 550 nm

[2] see Appendix A.4 for details on this well documented event

[3] More typical of average spring to-autumn-local DODs if one excludes the springtime Asian-dust dominated DODs (see Appendix A.3.2)

(0.0023 × 1.2$^{\pm1}$) would, assuming approximate proportionality, produce generally undetectable $BTD_{11\text{-}12}$ changes ($|BTD_{11\text{-}12}|$ <~ 3 × 0.0023 = 0.007 °K).

As another source of independent support for the general weakness of Arctic DODs, we note that AeF's summertime DOD statistics are ~ DOD computations derived from the simulated local dust ("Arctic dust") polar map (Figure 1 of Kawai et al., 2023 [KA]). The KA multi-year (2010-2019) "annual-mean vertically integrated mass concentrations" ("Arctic dust mass"

with units of mg-m$^{-2}$) yield DODs that ~ AeF's summertime CM AOD (see Figure A1 of Appendix A.3.2 where we compare the KA DOD simulations for the four AeF sites in or near the CAA). The summertime constraint on their comparison comes from the Asian dust domination of AeF's springtime CM AOD (Asian dust is a dust component that is not modelled by the KA simulations).

Having argued that Arctic DODs are, in general, at the margins of RS detectability, we must also take issue with B&V's

affirmation that: "While it is possible that a substance other than mineral dust is causing large-scale negative $BTD_{11\text{–}12}$ signatures in the polar environment, there is nothing in the literature to support this conjecture.". We will present an alternative mechanism below involving inversion-layer liquid (droplet) clouds whose cloud optical depths (CODs) are sufficiently large to induce significant negative $BTD_{11\text{–}12}$ signatures.

In general, there is often a tendency in the literature to significantly overestimate DOD magnitudes of Asian dust. It is no trivial

matter to decouple such relatively small DODs from very large CM CODs of clouds that are often in the neighbourhood of those dust plumes. Such clouds may indeed result from dust nucleation: see for e.g., Hildner et al. (2010) and their discussion involving a high-altitude Asian dust plume that apparently nucleates into a highly-depolarizing cloud (captured by the AHSRL lidar above our PEARL [Polar Environment Atmospheric Research Lab] observatory at Eureka, Nunavut). In more general terms, Eck et al. (2009) noted the shortcomings in applying the AERONET (temporally-based) V2 cloud screening algorithm

in the presence of spatially homogeneous clouds at Barrow, AK: spatially homogeneous (insufficiently variable) clouds such as thin cirrus are erroneously classified as dust (false-positive "dust"). More recently, Stone et al. (2014) underscored the potential for the same false-positive problem in their Barrow-based climatology of Arctic aerosols. Ranjbar et al. (2022) argued that the authors of a case study involving the transport of Asian dust into the high Arctic likely confused DOD with nearby COD and thereby significantly overestimated the DOD of a thin Asian dust plume about 7 km above the PEARL observatory.

Analogous problems plague polar winter data. O'Neill et al. (2016) used lidar profiles and a spectral cloud-screening approach to estimate the large (starphotometer-derived) CM AOD errors that would be associated with the application of frequently inadequate (temporally-based) cloud screening paradigms to polar winter optical depths acquired at the OPAL, Eureka site. The authors concluded that: "Spatially homogeneous clouds and low altitude ice clouds that remain after temporal cloud screening represent an inevitable systematic error in the estimation of AOD [more so for CM AOD]: the [positive bias] AOD

error was estimated to vary from 78 to 210% at Eureka and from 2 to 157% at Ny-Ålesund". In a not unrelated finding, Zamora et al. (2022) pointed out that the CALIPSO (CALIOP) classification algorithm was likely misclassifying wintertime "diamond dust" as mineral dust in their pan-Arctic analysis.

In terms of satellite-based estimates of DOD, B&V claimed that average AOD was "a proxy for dust aerosol concentration" and employed the 1998 to 2010 SeaWiFS AOD climatology of Hsu et al. (2012) to report a slight increase in AOD over the global oceans (and given their dust proxy assertion, a slight increase in DOD) in an apparent effort to support an increasing trend in their average RSED ("relative spatial extent of dust") parameter over the Arctic and Antarctic. This is yet another instance of DOD overestimation in the literature: AOD is almost universally dominated by fine mode (roughly submicron) particles and cannot be viewed as a proxy for "dust aerosol concentration" (while there is evidence that fine mode dust exists, there is little evidence that it dominates other types of fine mode aerosols). The proxy assumption is especially questionable when claiming to report a trend involving a minor AOD species (dust) coupled with a satellite AOD product that is less accurate than the AOD generated from ground-based AERONET data (for which a DOD trend analysis would be a challenge on its own merits): the bias error (amplitude $>\sim 0.01$) between the SeaWiFS AODs and AERONET AODs (Figure 2 of Hsu et al., 2012) (for example) is $>\sim$ the spread of AeFs spring and summer geometric standard deviation envelope for Resolute Bay $(0.0051 \times 3.0^{\pm 1})$.

A more realistic DOD satellite product over the Arctic is the MIDAS (ModIs Dust AeroSol) data set (Gkikas et al., 2021). The MIDAS reanalysis system is based on MODIS AODs coupled with MDFs (mineral dust fraction; a semi-intensive parameter of DOD $\div$ AOD) derived from MERRA-2 (Modern-Era Retrospective analysis for Research and Applications version 2) whose key components are the Goddard GOCART aerosol model and GEOS (Goddard Earth Observing System). The MIDAS annual DOD (arithmetic) mean for the 2003 – 2017 period (for the more accurate retrievals over water around Resolute Bay) is $\sim 0.01 \pm 0.02$ (where 0.02 is a computed estimate of retrieval uncertainty rather than a standard deviation). The AeF Resolute Bay value of $0.0051 \times 3.0^{\pm 1}$ is contained with the MIDAS uncertainty envelope.

## 2 Arctic aerosol events that are readily detected by remote sensing

Arctic aerosol events that are detectable using visible to near-IR, passive, satellite-based RS techniques are, for the most part, either FM (fine mode) smoke or, to a lesser extent FM Arctic haze. Xian et al. (2022) present a comprehensive pan-Arctic investigation of FM and CM AODs using reanalysis simulations of three aerosol transport models tied to satellite-based retrievals and verified (at the FM and CM AOD level) using a network of Arctic-AERONET stations. The monthly binned MRC (Multi-Reanalysis Consensus) AODs of their three models are shown as a function of aerosol species for the period of 2003 to 2019 (their Figure 2). The results for Resolute Bay show a year-round dominance of FM smoke and/or FM "ABF" aerosol (essentially anthropogenic sulphate-based Arctic haze or FM aerosols of biogenic origin) with CM dust aerosols having their greatest minority impact during the springtime Asian dust event (monthly arithmetic means of $\langle DOD \rangle < \sim 0.03$ compared to smoke and ABF monthly means of $\langle FM\ AOD \rangle < \sim 0.1$).

Returning to an event level case presented by VCT, the claim of a "strong dust event" associated with VCT's Figure 5 (and Figure 3b) imagery was (if aerosols were to be ascribed any role) associated with a FM smoke event induced by fires in Alaska and the Canadian Northwest Territories (see Figure S1a and its discussion). Figure S1b shows, if anything, that there is

marginal correspondence at best between the position of the smoke plume over the Amundsen Gulf (as evidenced by the pattern of the smoke on the true color image) and the patterns of negative blue colored $BTD_{11-12}$ values over the water regions south of Banks Island (at the southern extreme of the CAA). The principle optical effect in the massive region of blue-colored $BTD_{11-12}$ values to the west of Banks Island is largely associated with the presence of "liquid water" clouds or "uncertain" phase clouds (see the Worldview classifications of Figure S1b).

### 3 Negative BTDs associated with liquid phase clouds in the inversion layer

The spectral properties of water clouds, for CODs that are typically >> than the weak DODs described above, will likely dominate the $BTD_{11-12}$ spectral signature of Asian dust or local dust that is not within the immediate range of its drainage basin source. We found numerous examples of the presence of low-level CM water-clouds characterized by strong COD and weakly to strongly negative $BTD_{11-12}$ values over the Beaufort Sea (illustrated by the COD >~ 5 and $BTD_{11-12}$ >~ -1 °K, May 29, 2005 case study of Figures S2 to S5). Given the arguments presented above on the general weakness of DODs and the likely absence of any strong local dust source in the middle of the frozen Beaufort Sea, it is very unlikely that the massive region of negative $BTD_{11-12}$ values seen in cases such as that of Figure S3 could be attributed to the direct thermal influence of dust aerosols.

We found (over a 2011-2018 sampling period) persistent if irregular winter to spring (October to April) and summertime events of moderately negative MODIS $BTD_{11-12}$ values (-0.3 >~ $BTD_{11-12}$ >~ -0.8 °K) acquired near Barrow, where ground-based lidar and radar profiles indicated strong, super-unity CODs associated with physically thin, near-surface water clouds (see the Supplementary material "BTDBarrowSummer.xlsx" and "BTDBarrowWinter.xlsx" for details on our analysis of $BTD_{11-12}$ results near Barrow). Such low-altitude mixed-phase (water mixed with ice) clouds have been reported in the literature: de Boers et al. (2009) and Shupe et al. (2015) provide lidar / radar supported illustrations of mixed-phase events at Eureka and Barrow respectively[4]. The former paper reported a 4-year (2004-2007) frequency-of-occurrence (%) series (three-month-wide bins) of combined Barrow and Eureka results showing a general predominance of "SON" autumn bins (~ 10%) at <~ 1.5 km mean cloud-base-height[5] for single-layer, mixed-phase stratiform clouds. The latter paper provided a 2-year Barrow climatology (which is more relevant to the Barrow-region focus of the analyses that follows) showing that the monthly occurrence (%) was highest in October (~ 40%) while being only moderate and of lower altitude from March (~ 10%) to ~ 25% in April and May (with a strong preponderance of sub-1-km liquid occurrence). Yi et al., 2019 reported comprehensive satellite-based (MODIS and CALIOP) water cloud ("Arctic fog"), March-to-October statistics for a large Arctic Ocean region roughly centered north of Barrow. Their water cloud limitation to fog (cloud base height = cloud height – cloud thickness being < 1000 feet [300 meters]) was, however, rather restrictive with respect to the types of liquid cloud events that we

---

[4] The reader will note that for the purposes of this paper we do not distinguish between liquid clouds and mixed phase clouds. Below we argue that the ice COD in mixed phase clouds is typically negligible compared to the liquid water COD

[5] With a plume thickness of <~ 600 m. These are the 3-year results for Eureka: the single Barrow year of 2004, with a SON occurrence of 26% was more coherent with the Shupe et al. results.

investigate below (events requiring that the water cloud be imbedded in significant temperature slices of the Arctic inversion layer).

Nearly all our negative $BTD_{11-12}$ Barrow examples shared one feature that is rarely mentioned in typical $BTD_{11-12}$ literature; the ubiquitous and strong Arctic temperature-inversion up to altitudes ~ 1 km that occurs during the polar winter and summer (see, again, the Supplementary material "BTDBarrowSummer.xlsx" and "BTDBarrowWinter.xlsx" for details on our analysis near Barrow and Bradley et al., 1993 and Palo et al., 2017 for statistical summaries of the Arctic inversion layer). Inversion-layer cloud events are the most easily detectable instances of a fundamental principle; that a "warm-cloud" in the Arctic

inversion layer (in an atmosphere clear of higher altitude clouds) transforms the more common negative-lapse-rate $BTD_{11-12}$ signature from a generally positive to negative dependency with the degree of negativity being dependent (for a given surface emissivity, inversion-layer strength and water vapour load) on the COD and effective particle radius (see, for example, Key, 2002 and Liu et al., 2004).

We generated MODTRAN-simulated $BTD_{11-12}$ vs $BT_{11}$ graphics whose input parameters encompassed a wide variety of COD

and particle size conditions about a specific March 22, 2015 event at Barrow (see the discussion of that event associated with Fig. 1 below). The resulting $BTD_{11-12}$ vs $BT_{11}$ patterns are shown in Figure B1 while Table 1 below presents a descriptive summary of those simulations[6] (the optical details and boundary layer conditions associated with each $BTD_{11-12}$ vs $BT_{11}$ pattern are given in Appendix B.1). The "convex downward" shape of a large-COD water cloud in an inversion layer will produce almost exclusively negative values that fundamentally depend on the COD = 0 and ∞ singularities on the $BTD_{11-12}$ vs $BT_{11}$

patterns of Figure B1 (while a high-altitude ice or liquid cloud will produce, as per the upper left graphic of Figure B1, generally positive $BTD_{11-12}$ values).

Table 1 - Empirical (E) & simulated (S) $BTD_{11-12}$ vs $BT_{11}$ results for a variety of cloud or dust plumes embedded in positive temperature-lapse-rate (inversion layer) or negative lapse-rate regions. Blue, red and brown text refer respectively to water clouds, ice clouds and dust plumes.

| Temperature lapse rate ($dT/dz$) | Emissivity Slope ($d\varepsilon/d\lambda$) | $BTD_{11-12}$ vs $BT_{11}$ pattern | Cloud or dust plume & surface scenarios | Citations / comments |
|---|---|---|---|---|
| Negative | Negative | Convex upwards[1] | Low-altitude to mid-altitude water cloud | Baum et al. (2000) (S) [a], Key (2002) (S) [b] |
| Positive | Negative | Convex downwards[2] | Low-altitude (inversion layer) water clouds | This study[c] (S & E) Key (2002) [d] (S) |
| Negative | Negative | Convex upwards | High altitude ice clouds | This study (S[e] & E). |
| Positive | Negative | Convex downwards | Low-altitude (inversion layer) ice clouds | This study (S[f]) |
| Negative | Positive | Convex downwards | High altitude Asian dust plumes | Various (S & E) This study (S)[h] |
| Positive | Negative | Convex upwards | Low altitude (inversion layer) Asian dust plume parameters | This study (S) |

[1] Otherwise known as concave downward. These curves generally (but not always) consist of positive $BTD_{11-12}$ values

---

[6] as well a simulations and empirical evidence from the literature

[2] Otherwise known as concave upward. These curves generally (but not always) consist of negative $BTD_{11-12}$ values

[a] Surface to top-of-water-cloud temperature decrease of $\cong 297$ to $\cong 284$ °K (low altitude water cloud)

[b] Surface to top-of-water-cloud temperature decrease of 263.16 to 238.65 °K (mid altitude water cloud)

[c] Surface, top-of-inversion-layer-cloud and top of high-altitude cloud temperatures $\cong 256$, 262 and 213 °K. See the captions of Figures S6 and the discussion of Figure B1 for details

[d] Surface to top-of-water-cloud temperature increase of 235.66 to 253.15 °K

[e] Same approximate values as footnote c. The high altitude ice-cloud literature is resplendent with convex-upward $BTD_{11-12}$ vs $BT_{11}$ examples (see, for example, Figure 3 of the tropical atmosphere simulation of Ackerman et al., 2010)

[f] Same approximate values as footnote c

[g] There are many **S** & **E** examples in the literature of convex-downward $BTD_{11-12}$ vs $BT_{11}$ patterns (see, for e.g., Figures 2 and 4 of Zhang et al., 2006). The generally negative nature of those patterns are a well-known high altitude signature of Asian dust plumes.

Table glossary

**E** – empirical results, **S** – simulated results

$\varepsilon$ – emissivity spectrum of a surface composed of water, ice or dust soil

$dk/d\lambda$ & $d\varepsilon/d\lambda$ – 11 to 12 µm spectral slope of respectively, the complex part of the refractive index & emissivity

Figure 2a shows a radar profile for the specific inversion-layer (Figure S6) Barrow illustration of the March 22, 2015 event (with Figure 2b providing a zoom of the radar profile in the inversion layer between 0 and 2 km). Figure 2c shows both the MODIS-measured moderate-amplitude $BTD_{11-12}$ values and the MODTRAN-simulated values (details in the figure caption). This demonstrates how (i) a warm, liquid-water, inversion-layer (negative $BTD_{11-12}$) cloud (whose lower and upper boundaries

are explicitly shown in Figure S6 and Figure 2b) coupled with the positive $BTD_{11-12}$ presence of a cold (negative or normal lapse rate) ice-cloud around 6-km-altitude (during roughly the first half of the displayed time period) produces systematically varying $BTD_{11-12}$ values that oscillate between the negative to positive extremes of the two phenomena. (ii) that MODTRAN radiative transfer simulations were largely successful in capturing the $BTD_{11-12}$ oscillations.

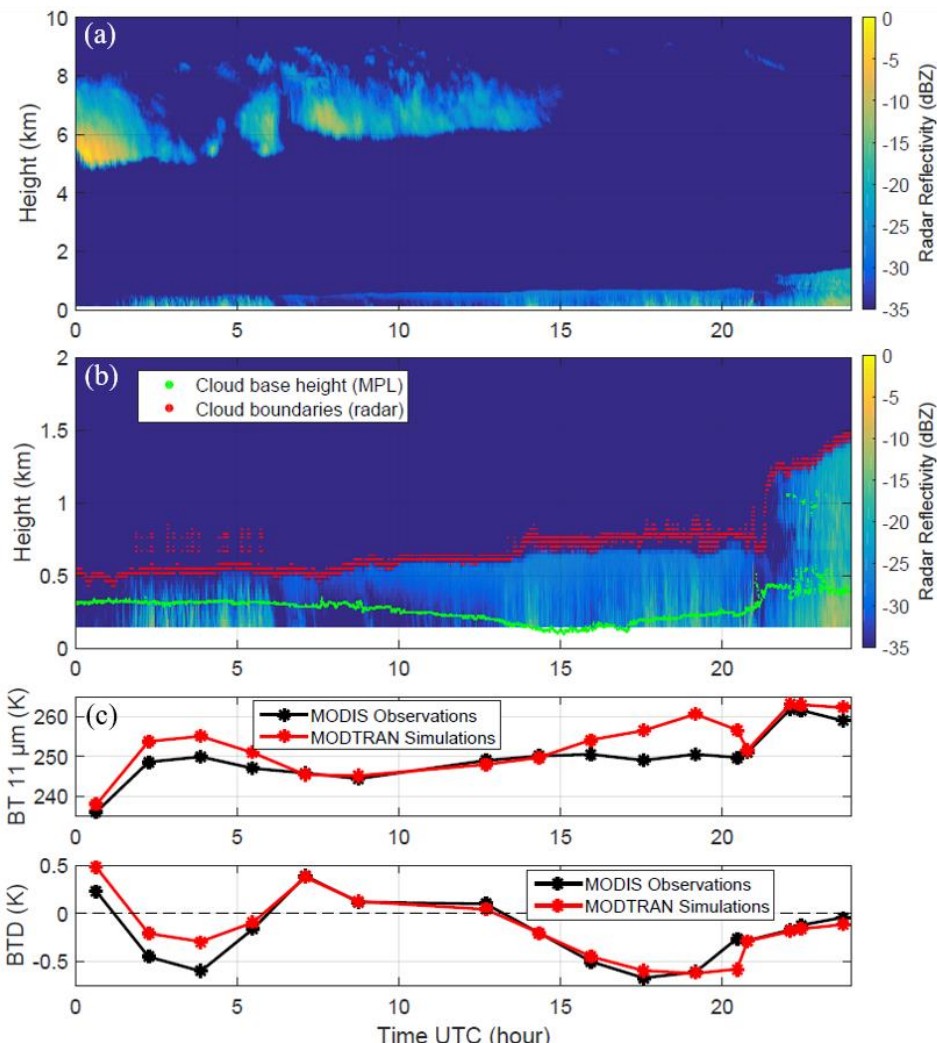

Figure 2 – Radar backscatter coefficient profiles acquired at the ARM Barrow site on Mar. 22, 2015: (a) 0 to 10 km altitude and (b) a zoom from 0 to 2 km altitude. The green and red curves (as measured by the combination of lidar and radar backscatter profiles) indicate, respectively, the bottom and top of what we inferred to be a water-dominated, mixed-phase cloud that was partially contained in an inversion layer. The dynamical details of this event, including radiosonde temperature profiles, are given in Figure S6 and its caption. The solid black curves of Figure (c) show, for the MODIS pixel nearest to the Barrow site, the MODIS-measured $BT_{11}$ time series (upper graph) and MODIS-measured $BTD_{11-12}$ time series (lower graph). The solid red curves show their MODTRAN-simulated analogues. These simulations employed input parameters representative of that day (see the captions of Figure S6 and S7 for further details).

Specific details on the vertical extent of the mixed phase backscatter coefficient profile, the water vs ice COD contributions and their relationship to the temperature (inversion layer) profile provided by the Barrow radiosonde profiles are presented in the discussion of Figure S6. In those details, we argue that the radar profiles provide key information about the upper boundary of the water / mixed phase cloud (beyond the upper bound defined by the extinction limit of the lidar) and its attendant extension to altitudes where there was actually a strong temperature inversion.

The same (Table 1) inversion layer $BTD_{11-12}$ convexity reversal should apply to warm, low-level ice-clouds in an Arctic inversion layer: however as shown in Figure B1(b), the amplitude of the convex downward pattern can be insufficient to move the pattern into the negative $BTD_{11-12}$ range. In any case, we did not, in our survey of significantly negative, near-Barrow $BTD_{11-12}$ (MODIS) values, find any obvious lidar / radar retrievals dominated by synchronous, low-level, optically-thick ice-clouds in the inversion layer. This is not unexpected since the CODs of near-surface ice clouds are substantially smaller than those of water clouds (see Shonk et al., 2019 for a general statement and specific examples in the Dec. 29, 2006 Eureka case study of de Boer et al., 2009 as well as Sections 3d vs 4c of Zuidema et al., 2005 for a May 1 – 18, 1998 case study at a floating ice camp ~ 600 km northwest of Barrow). Morrison et al. (2012) also point out the dominance of water CODs over ice CODs across a 5-day (May 11 – 15, 2011) Eureka event and underscore the persistence of Arctic mixed-phase clouds in general. This dominance of water COD over ice COD was also found in our Figure 2 case study (see the caption of Figure S6 for details). We would note that a convex-down to convex-up pattern reversal would occur when comparing high altitude dust clouds with warm dust clouds located in an Arctic inversion layer: the generally negative contours for the former would be transformed to positive contours (see Figure B1(d)) that would actually confound the classic negative $BTD_{11-12}$ signature of cold, high-altitude dust clouds (see for example, Figures 3 (MODIS $BTD_{11-12}$) and Figure 4 (simulations) of Zhang et al., 2006). Support for this affirmation for the case of local dust comes indirectly from Ranjbar et al. (2021): the lapse rate in the Lake Hazen case was, in all likelihood, an (inversion-layer-free) rate of decreasing temperature with increasing altitude. This results in the negative $BTD_{11-12}$ plume values reported in that paper (the defining $BTD_{11-12}$ vs $BTD_{11}$ pattern is more in the nature of the convex-down shapes of Figure B1(c)).

The overarching message of this section is that negative $BTD_{11-12}$ values in the Arctic are not a unique signature of the pervasiveness of dust across the Arctic. The $BTD_{11-12}$ signature of airborne dust in the inversion layer would generally be too weak to detect and of the wrong sign with respect to the classical negative signature of desert dust plumes in a normal lapse rate environment. DODs in the Arctic are generally too small to induce significant $BTD_{11-12}$ amplitudes. The $BTD_{11-12}$ signature of deposited dust can be significantly negative but, as suggested by our case study on snow deposition of local dust (see the following section), tend to be spatially limited to dust dominated regions of the drainage basin source.

## 4 RS detectability of dust impacts

An affirmation of the general marginality of airborne-dust RS detectability in the Arctic is not to say that the impacts of Asian or local dust are necessarily marginal in terms of satellite-based RS. The cumulative deposition of local dust associated with weak DODs can (as also noted by VCT) be substantial over seasonal or longer time scales with significant changes in surface reflectance (and attendant impacts on early snow melt coupled with a feedback effect of even greater reflectance changes). AVHRR remote sensing imagery dating back as far as 1991 was employed by Woo et al. (1991) to argue that dust covered areas on the Fosheim Peninsula (region of Eureka) were the first to experience snow melt (Ranjbar et al., 2021 showed image browning regions on MODIS "Corrected Reflectance (True Color)" (RGB) images that corresponded to Woo's "dark spot"

regions). Figure S8 shows, what we argue, are examples of dust deposition on snow or ice in the neighbourhood of southern-CAA drainage basins whose flow dynamics have induced local dust plumes. This illustrates how the accumulation of local dust deposition by dust plumes produces, (i) true-color images of significantly modified snow reflectance in the visible spectral region (<~ 60% average reflectance change[7]) but only (ii) weakly positive $BTD_{11-12}$ dust-deposition signatures near their drainage basin sources (while also showing that significantly negative $BTD_{11-12}$ signatures do occur in what are likely the very localized pure dust regions of the drainage basins). This, as indicated in the legend of Figure S8, is likely a $BTD_{11-12}$ difference resulting from the greater dust-surface emissivity of band 12 relative to band 11. These illustrations strongly suggest that, significantly negative signatures of dust on snow or ice are likely to be very limited in their spatial extent.

The reflectance effects associated with the deposition of Asian dust on snow are less evident. Asian dust deposition was detected by ground teams at higher altitudes (where sources of local dust would be unlikely) in the Mount Logan (Yukon) region of the St. Elias range during the strong April 2001 Asian dust event (Zdanowicz et al., 2006). The authors suggested that up to 45% of the airborne dust mass abundance was deposited in the snow (over a 9 day period) and that the mechanism for deposition was scavenging by snow flakes. MODIS RGB images show no obvious impact: this effort to determine an impact is not helped by these agents of deposition arguably confounding / camouflaging the darkening impact of dust. Zhao et al. (2022) employed a variety of ground- and satellite-based, passive and active RS data as well as surface nephelometer measurements of CM scattering coefficient to investigate the albedo (spectrally integrated reflectance) impact of dust deposition on snow during March 14, 2013 and April 20, 2015 Asian dust events over Alert (Nunavut) and Barrow, Alaska respectively[8]. The CM scattering coefficients coupled with estimates of the dust plume (mid-tropospheric) altitude over each site suggested direct deposition links between the dust plumes and the surface dust (they did not attempt to elaborate on any explanation of the deposition dynamics). The authors employed a radiative transfer model to argue that daily dust deposition events could reduce snow surface (panchromatic) albedo by as much as 2.3% at Barrow and 1.9% at Alert. These albedo reductions would be quite substantial if dust depositions (in relatively unperturbed snow conditions) were allowed to accumulate over, for example the 9-day period of the April 2001 event. However, the simulations of Groot Zwaaftink et al. (2016) on the substantially greater contribution of local Arctic dust (versus Asian or African dust) to dust deposition suggests that Asian and African dust would, in general, play a secondary reflectance perturbation role compared to local dust.

A second substantial impact of Arctic dust particles is associated with their role as INP (ice nucleation particles) and their indirect effect on cloud dynamics. The core message of Kawai et al. (2023) was not a statement about the weak optical influence of local Arctic dust, but rather a simulation-based affirmation that local dust was the dominant INP source in the lower Arctic troposphere during summer and fall. A similar statement concerning the dominance of local dust over Asian dust as INPs was made by Xi et al. (2022) based on INP (droplet freezing) measurements made near the source of local dust plumes at the sub-Arctic Lhù'ààn Mân' (Kluane Lake) site in the Canadian Yukon territory. Barr et al. (2023) reported on the greater INP activity

---

[7] c.f. the "Dusty Snow" reflectance changes in the Figure 1a spectra of Painter et al., 2007

[8] those two intensive-analysis days were supplemented by neighbouring days for which CM nephelometer measurements suggested the dust-event extended beyond those two core days.

of local dust (from glacial drainage basins on the southern Alaskan coast) relative to Arctic dust from low-latitude sources. Tobo et al. (2019) described the important role of local dust as an INP source in the Svalbard region and noted that the high ice nucleating ability of the local dust was likely governed by the presence of organic matter. Shi et al. (2022) analyzed the radiative forcing impacts of local Arctic dust (what they called "HLD" for high-latitude dust whose source is in the Arctic or sub-Arctic). Their simulations (roughly supported by INP comparisons with measured INPs carried out at 9 stations) show,

for example, that HLD INPs likely instigated a maximum depletion in the liquid water path (LWP) of mixed phase clouds in the fall season (and lesser but still significant LWP changes during the summer and winter seasons). Those LWP depletions ($\sim 8$ g-m$^{-2}$) amount to water COD reductions of $<\sim 1.5$ (at any wavelength for which the Mie extinction efficiency (Q) is $\sim$ the optically large-particle asymptotic value of $\sim 2$). Such COD changes (along with their associated extinction coefficient profile change) would be readily detected using standard passive & active, satellite-based sensors (from the visible to the thermal IR).

**5 Conclusions**

We presented a variety of examples showing how direct RS-based estimates of CM Arctic dust were oftentimes excessively large due to a failure in separating out the contribution of CM clouds (or cloud-like optical contributions). A particular emphasis was placed on a paper by Vincent (2018) who reported an optically strong airborne dust presence in the western Canadian Arctic that was ascribed to dust of Asian origin or dust from local sources. While we do not dispute the presence of

290 both Asian and local dust in the Arctic, the direct RS detectability of airborne dust (attributed to satellite-measured values of significantly negative BTD$_{11-12}$ values) was almost surely misrepresented. While it is difficult to account for all examples of strongly negative BTD$_{11-12}$, it is very unlikely that airborne dust plays a major RS role in any case other than plumes of strong DOD ($> \sim 0.5$). One, much more likely contributor would be water clouds (or, more generally stated, water dominated, mixed phase clouds) in the Arctic inversion layer.

The RS detectability of the impact of Arctic dust and notably Arctic dust from local drainage basin sources can, however, be of significance. Sustained dust deposition can substantially decrease the (visible to shortwave IR) snow and ice reflectance and the attendant signal measured by satellite sensors (while significantly negative BTD$_{11-12}$ values represent an extremely limited area according to our event level case studies). The substantial INP (Ice Nucleating Particle) role of local Arctic dust can, for example, induce significant changes in the properties of low-level mixed phase clouds (optical depth changes $<\sim$ unity) that

can be readily detected by active and passive RS instruments. It is clearly critical that the distinction between the RS detectability of Arctic dust versus the RS detectability of the impacts of Arctic dust be understood if we are to properly account for and model the radiative forcing impacts of dust in the climate sensitive Arctic region.

## 6 Appendices

## Appendix A – Intensive and extensive microphysical and optical parameters of local and Asian dust

### A.1 Effective radius relationships for spherical particles

The effective radius for spherical particles is defined by Hansen & Travis (1974) (HT) as:

$$r_{eff} = \frac{\int r^3 \frac{dn}{d\ln r} d\ln r}{\int r^2 \frac{dn}{d\ln r} d\ln r} = \frac{\int (D/2)^3 \frac{dn}{d\ln D} d\ln D}{\int (D/2)^2 \frac{dn}{d\ln D} d\ln D} = \frac{1}{2} \frac{\int D^3 \frac{dn}{d\ln D} d\ln D}{\int D^2 \frac{dn}{d\ln D} d\ln D} = \frac{1}{2} D_{eff} \qquad (A1)$$

where the very last relation amounts to a definition of $D_{eff}$. Equation (A1) can then be recast in terms of total particle-surface and particle-volume concentration:

$$D_{eff} = \frac{\frac{2^3}{\frac{4}{3}\pi} \int \frac{4}{3}\pi \left(\frac{D}{2}\right)^3 \frac{dn}{d\ln D} d\ln D}{\frac{2^2}{\pi} \int \pi \left(\frac{D}{2}\right)^2 \frac{dn}{d\ln D} d\ln D} = \frac{3 \int \frac{dv}{d\ln D} d\ln D}{2 \int \frac{ds}{d\ln D} d\ln D} = \frac{3}{2} \frac{v}{s} \qquad (A2a)$$

From equation (A1) the effective diameter can be recast as:

$$D_{eff} = \frac{3}{2} \frac{\int \frac{4}{3}\left(\frac{D}{2}\right) \frac{ds}{d\ln D} d\ln D}{\int \frac{ds}{d\ln D} d\ln D} = \frac{\int D \frac{ds}{d\ln D} d\ln D}{\int \frac{ds}{d\ln D} d\ln D} = \frac{\int D \frac{ds}{d\ln D} d\ln D}{\int \frac{ds}{d\ln D} d\ln D} = \langle D \rangle_{\omega = ds/d\ln D} \qquad (A2b)$$

the weighted mean of $D$ where the weight $\omega = ds/d\ln D$. Ginoux (2003) argues that the shape of dust particles are, in general, better represented by prolate ellipsoids (see the following section).

### A.2 Computation of $D_{eff}$

Kawai et al. (2023) (KA) employed Kok's particle-volume size distribution as the basis of their multi-year simulations (ultimately it was the starting point[9] of their computations of seasonally averaged particle-mass columnar densities). Kok's particle-volume size distribution (his equation (6)) is related to his particle-number size distribution (his equation (5) by $dV_d/dlnD_d = C_N/C_V D_d^3 dN_d/dlnD_d$. We can[10] recast this as;

$$d\tilde{v}/dlnD = C_N/C_V [3/(4\pi) 2^3] [(4/3)\pi (D/2)^3] d\tilde{n}/dlnD$$
$$= (C_N/C_V 6/\pi) v_{sp}(D) d\tilde{n}/dlnD$$
$$= C_{Kok} v_{sp}(D) d\tilde{n}/dlnD \qquad (A3)$$

where $v_{sp}(D) = (4/3)\pi (D/2)^3$ is the volume of a spherical particle of radius $D/2$ and $C_{Kok} = C_N/C_V \times 6/\pi$. However $C_{Kok}$ (= 0.144 for Kok's $C_N$ and $C_V$ values of 0.9539 and 12.62 µm respectively) is not close to unity as would be expected for

---

[9] The emission (source) particle-volume size distribution
[10] dropping his "d" (dust) subscript, using lower case letters for these point-volume parameters and $\tilde{n}$ and $\tilde{v}$ for their "normalized" distributions

small dust particles. Equation (A3) is apparently the correct inter-distribution relationship between Kok's (Figure 6) "normalized" number and "normalized" volume size distributions[11].

However equation (A2a) applied to Kok's normalized distributions gives unrealistic estimates of the effective radius (0.78 µm)[12]. Those normalized distributions were not defined and so we have to tentatively conclude that the normalization precluded the application of equation (A2a)[13]. Dust particles are not sufficiently large to have substantial non-sphericity effects and so

one expects the departure of $C_{Kok}$ from unity to be relatively small. Ginoux (2003) cited Okada et al. (2001) to indicate that dust particles near their source (Chinese desert sites) displayed an ellipsoid aspect ratio (λ) of ~ 1.5 and that moderately higher values of 2 showed no significant departure from sphericity (their Figure 5, for example, shows that simulated particle volume distributions for λ = 2 were quite close to the spherical-particle AERONET inversions for 6 sites near or in the desert sources of dust). Accordingly we can, in general, treat dust particles as being approximately spherical ($d\tilde{v}/dlnD \sim v_{sp}(D)\, d\tilde{n}/dlnD$)

and the light-grey broken-line open-circles in the Supplementary material ("Local_dust_PSDs.xlsx") represent the appropriate distribution[14] for the employment of equation (A2a). This yields a $D_{eff}$ value of 5.40 µm ($r_{eff} = 2.70$ µm).

## A.3 DOD computations for KA's local dust particles

### A.3.1 $DOD$ mass efficiency ($DOD_m$)

If $V$ is the columnar, particle-volume abundance $\rho$ is the dust particle density and $A$ is the particle-number abundance then the

particle-mass abundance ($A_m$) in the case of the KA local dust simulations[15] (or any unimodal particle-volume or particle-mass distribution) is given by;

$$A_m = \rho V \cong \rho \frac{4}{3} \pi r_{eff}^3 A \qquad (A4a)$$

$$A \cong \frac{A_m}{\rho \frac{4}{3} \pi r_{eff}^3} \cong \frac{A_m}{m} \qquad (A4b)$$

where the concept of intensive parameters averaged over a unimodal particle size distribution is discussed, for e.g., in O'Neill

et al. (2005). If the dust extinction cross section is $\sigma$ and Q is the dust extinction efficiency, then the dust optical depth (DOD) is;

$$\tau \cong \sigma A \cong \sigma \frac{A_m}{\rho \frac{4}{3} \pi r_{eff}^3} = Q \pi r_{eff}^2 \frac{A_m}{\rho \frac{4}{3} \pi r_{eff}^3} = \frac{Q\, A_m}{\frac{4}{3} \rho\, r_{eff}} \qquad (A5)$$

---

[11] as verified by the fact that the black, solid-line, open-circle ($C_{Kok} v_{sp}(D)\, d\tilde{n}/dlnD$) curve is very close to Kok's gold, solid-line full-circle curve ($d\tilde{v}/dlnD$) in the Supplementary material ("Local_dust_PSDs.xlsx")
[12] Versus, for example, a volume-weighted geometric mean diameter (VMD) of 6.51 µm (the AERONET-inversion type of calculation)
[13] Meaning, that Kok's normalized distributions were not equally proportional to their physical representations (the physical representations being symbolized by the hatless variables in his section
[14] Meaning that we treat the distributions as being spherical-particle distributions
[15] The parameter that they call "vertically integrated … mass concentrations"

Employing the mean $r_{eff}$ of 2.7 μm from the Kok distributions (Appendix A.2 above) yields $x_{eff} = 2 \pi r_{eff}/\lambda = 33$ for $\lambda = 0.5$ μm. Q approaches an asymptote ~ 2.3 for values of the product $x_{eff}(1 - m_r) >\sim 10$ (Figure 16.3 of Hinds, 1999 with $m_r$ being the real part of the refractive index) and refractive indices representative of dust[16]. Employing the MITR[17] density of 2.6 g-cm$^{-3}$ $\rightarrow$ 2.6 × 10$^9$ mg/m$^3$ yields;

$$\tau \sim \frac{2.3 \, A_m}{\frac{4}{3}(2.6 \times 10^9) \, \text{mg/m}^3 \times (2.65 \times 10^{-6}) \, \text{m}} \sim 0.250 \times 10^{-3} \, (\text{mg/m}^2)^{-1} \, A_m \quad \text{(A6)}$$

One can define a "DOD mass efficiency" $\times 10^4$ (DOD per unit columnar mass abundance) as;

$$DOD_m \times 10^4 = \frac{\tau}{A_m} \sim 2.5 \, (\text{mg/m}^2)^{-1} \quad \text{(A7)}$$

### A.3.2 DOD extracted from KA's particle-mass abundances

KA's multi-year $A_m$ averages derived from their Figure 1 at the position of AeF's sites that are in or near the CAA, were employed to compute the DOD estimates of Table A.1 below. We note that "Eureka" is mean't to represent the similar environments of two Eureka sites (the sea-level OPAL site and the 610-meter elevation "PEARL" (Ridge lab) site): the resolution of KA's $A_m$ values allows no such distinction to be made for the KA simulations.

Table A1 – Local dust DODs derived from the $\boldsymbol{A_m}$ (mass abundances) of KA's Figure 1 (at the position of the four AeF sites within or near the CAA).

| Site | Lat., Long. | $A_m$ [mg/m$^2$] | Computed DOD ($\tau$) |
|---|---|---|---|
| Resolute Bay | 75°N, 95°W | 14.68 | 0.0037 |
| Eureka (PEARL & OPAL) | 80°N, 86°W | 0.32 | 0.000079 |
| Thule | 77°N, 69°W | 3.16 | 0.00079 |

Figure A1 shows a comparison between AeF's CM AOD (geometric mean) summertime (JJA) climatology and the KA simulations. The KA simulations include no Asian dust component: their yearly means should, in principle be ~ the summertime AeF values (if the latter can be assumed to be dominated by local dust). The AeF CM AODs of the April & May springtime period are (as per AeF), largely dominated by Asian dust with values >~ 0.06 (off-scale in Figure A1). We also note that PEARL and OPAL are separated by a distance of only 15 km with their elevations being, respectively, 610 and 10 m. The simple subtraction labelled "OPAL – PEARL" in Figure A1 is arguably a better measure of local dust DOD than their individual summertime means (assuming the local dust is largely limited to altitudes less than that of the PEARL (Ridge lab) site ). The KA precision for the Resolute Bay, Thule and "OPAL – PEARL" (Eureka) site would then nominally be ~ 0.001. This is a number that one hesitates to quote given the preponderance of uncertainties that plague both the simulations and the measurements (for e.g., the quality, respectively of local dust emissions about a given AeF site given the coarse KA spatial

---

[16] 1.53 – 0.0078i for the MITR ("Mineral transported") class of dust (the "opdat" directory of the OPAC package). Hess et al.'s MITR density transforms as 2.6 g/cm$^3$ = 2.6 × 10$^3$ mg/(10$^{-2}$ m)$^3$ = 2.6 × 10$^9$ mg/m$^3$

[17] Optical and μphysical parameters are listed in Table 1c of Hess et al. (1998).

resolution of 1.9° × 2.5° and the nominal AERONET AOD error ~ 0.01). The most optimistic affirmation is arguably that the summertime dust estimates are the same order of magnitude.


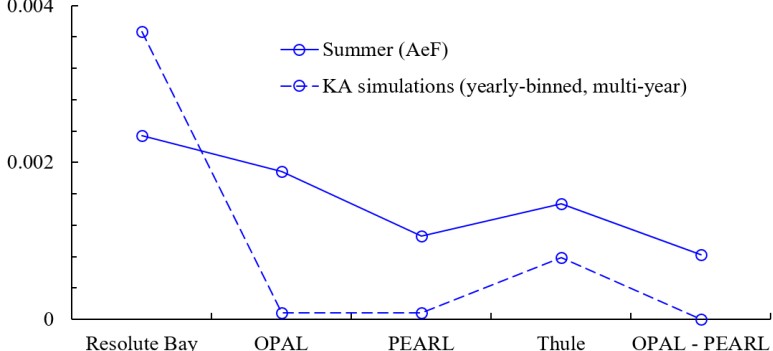

Figure A1 – Local dust DODs derived from the $A_m$ (mass abundances) of KA's Figure 1 (at the position of the four AeF (AERONET) sites in or near the CAA) along with summertime (JJA-averaged) CM AODs from AeF's four sites.

### A.3.3 KA-model "underestimation" of local DOD

Figure S4 of KA's supplementary material suggests that their simulated 550 nm "annual-mean zonally averaged dust AOD" significantly underestimates the local DOD relative to the CALIOP 532 nm estimate of local DOD[18]. The 3rd column of Table A2 (corresponding to the 2nd column of Table A1) shows the DODs ($\tau$) computed from KA's Figure 1 at AeF's AERONET sites. The 4th column is a "correction" of KA's DODs (DOD - ΔDOD) to yield values that would account for the gap between the KA DODs and the CALIOP DODs. The result, relative to AeF's summertime estimate is a better comparison for Eureka

and Thule and a rather large overestimate for Resolute Bay. The limitation of such a "correction" is the credibility of CALIOP in classifying and estimating local DOD in the Arctic. There is no validation in the literature of CALIOP's capability in classifying local dust plumes. The conclusion of this exercise is simply that there is no reason to change the order of magnitude conclusion of the previous section.

Table A2 – Computed DOD values (DOD ($\tau$)) employing the nominal $DOD_m$ value of equation (A7) and corrected DOD values (DOD -
ΔDOD) .

| Site | Lat., long. | Computed DOD | "Corrected DOD" (DOD - ΔDOD$^a$) |
|---|---|---|---|
| Resolute Bay | 75°N, 95°W | 0.0037 | 0.0087 |
| Eureka (PEARL & OPAL) | 80°N, 86°W | 0.000079 | 0.0031 |
| Thule | 77°N, 69°W | 0.00079 | 0.0038 |
| $^a$ The simulations underestimate the CALIOP "truth": their ΔDOD bias is accordingly negative | | | |

---

[18] Underestimates by a bias which we label ΔDOD. There are, for example, biases of ΔDOD ~ - 0.005 at the 75°N lat. of Resolute Bay and ~ - 0.003 at the near 80°N lat. of Eureka and Thule: these values were estimated from KA's Figure S4 ( (-1) × [CALIOP curve – red KA simulation curve] ).

## A.4 Estimation of Mount Logan DODs during the Asian dust event of April 2001

Table A3 shows visually extracted DODs from NAAPS simulations over the region of Mount Logan (Yukon) during the April 11 to 19, 2001 Asian dust event. The DOD values are the midpoints of the standard NAAPS color-scale bins. If there is no NAAPS DOD (no NAAPS dust at the position of Mount Logan) then the bin is assigned a value of 0.0[19]. The arithmetic average of all the DOD values below is $< DOD > = 0.13$. This table supports the discussion above surrounding the well documented Asian dust event of 2001 and the dust deposition consequences in the Mount Logan region.

Table A3 – Visually determined NAAPS DODs[20] at Mount Logan, YK (60° 34' N, 140° 24' W) during the April, 11 to 19, 2001 Asian dust event.

| ddhh (UT) | $DOD_{bin\ center}$ | ddhh (UT) | $DOD_{bin\ center}$ | ddhh (UT) | $DOD_{bin\ center}$ |
|---|---|---|---|---|---|
| 1100 | 0.0 | 1400 | 0.0 | 1700 | 0.15 |
| 1106 | 0.0 | 1406 | 0.15 | 1706 | 0.15 |
| 1112 | 0.0 | 1412 | 0.3 | 1712 | 0.15 |
| 1118 | 0.0 | 1418 | 0.3 | 1718 | 0.15 |
| 1200 | 0.0 | 1500 | 0.3 | 1800 | 0.15 |
| 1206 | 0.15 | 1506 | 0.3 | 1806 | 0.15 |
| 1212 | 0.15 | 1512 | 0.3 | 1812 | 0.0 |
| 1218 | 0.15 | 1518 | 0.3 | 1818 | 0.0 |
| 1300 | 0.15 | 1600 | 0.15 | 1900 | 0.15 |
| 1306 | 0.0 | 1606 | 0.15 | 1906 | 0.15 |
| 1312 | 0.15 | 1612 | 0.15 | 1912 | 0.0 |
| 1318 | 0.15 | 1618 | 0.15 | 1918 | 0.0 |


---

[19] or a value of 0.15 if the edge of the 0.15-valued (yellow) colored plume cannot be visually separated from the position of Mount Logan

[20] https://www.nrlmry.navy.mil/aerosol-bin/aerosol/display_directory_all_t.cgi?DIR=/web/aerosol/public_html/globaer/ops_01/noramer/200104&TYPE=

## Appendix B – Computational details in support of Table 1

### B.1 MODTRAN simulations of $BT_{11\text{-}12}$ vs $BT_{11}$ patterns for liquid water, ice and dust

Figure B1 shows MODTRAN $BTD_{11\text{-}12}$ vs $BT_{11}$ patterns over a "Snow/Ice" surface for water and ice clouds (left hand graphs) and dust plumes (right hand graphs) at high altitude (top graphs) and within a low-altitude inversion layer (bottom graphs). The general atmospheric conditions and cloud parameterization represent a range of values that include the specific conditions of Figures 2a, 2b, S6 and S7 (conditions of Barrow and its surroundings on March 22, 2015). The temperatures employed in the MODTRAN simulations at the snow/ice surface, inversion-layer-cloud-top and high-altitude cloud-top were, respectively 255.56, 262.05 and 212.66 °K. These graphs provide support for all the simulation (S) based $BT_{11\text{-}12}$ vs $BT_{11}$ classifications ($BTD_{11\text{-}12}$ vs $BT_{11}$ pattern characterization) of Table 1.


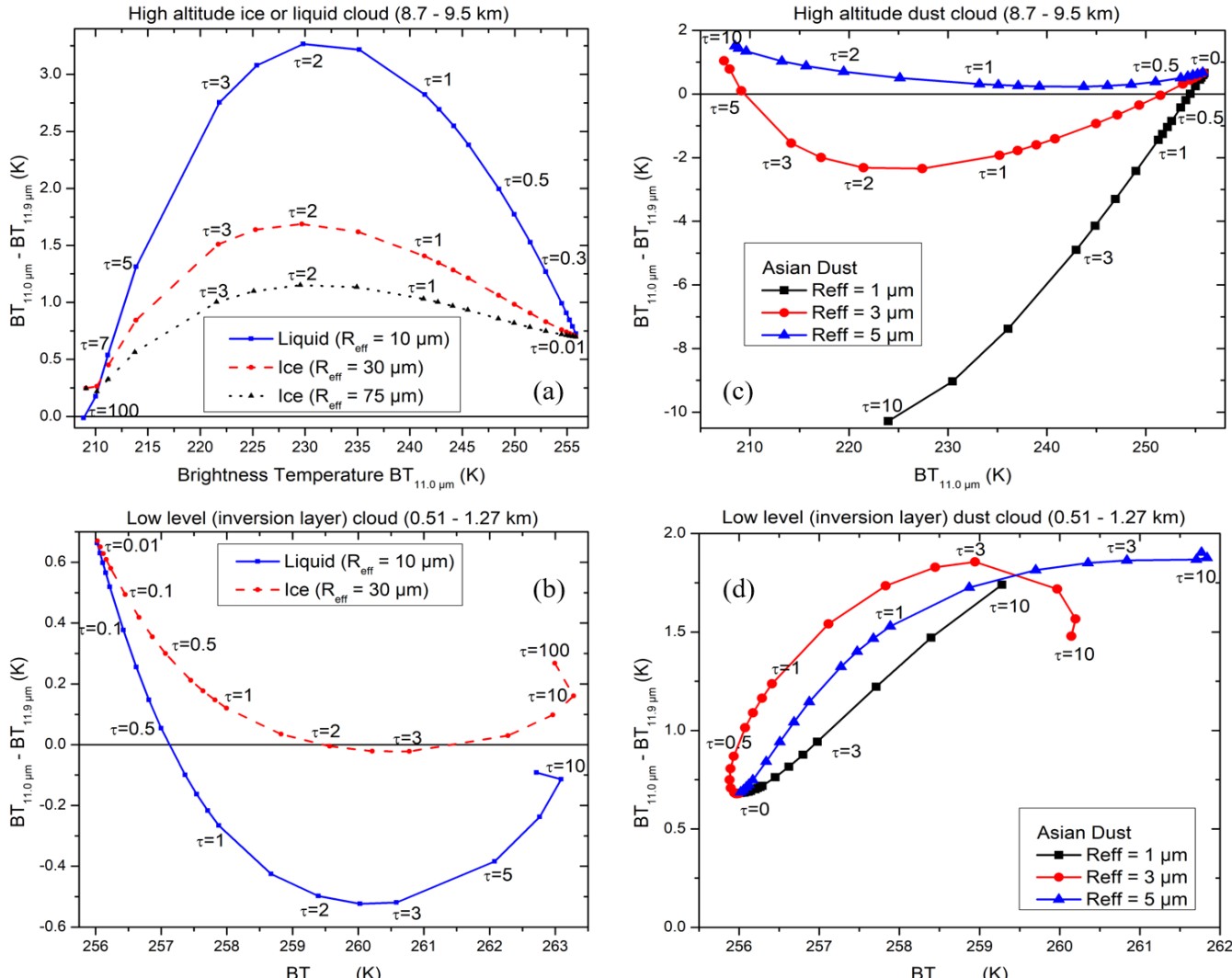


Figure B1 – $BTD_{11-12}$ vs $BT_{11}$ simulations for different types of clouds over a "Snow/Ice" surface (surface of Feldman, 2014). Graphs (a) and (b) respectively: ice and liquid cloud at a high altitude and within a low altitude temperature inversion layer. Graphs (c) and (d) respectively: Asian dust cloud at a high altitude and within a low altitude inversion layer. The wavelength-dependent optical depth ($\tau$) is reported at a wavelength of 550 nm in order to make a link with optical effects in the visible wavelength region. The significant curvature of the low level "Reff = 3 um" red curve in the bottom right graph of Figure B1 suggests a balanced radiative transfer condition wherein there is little change in $BTD_{11}$ with increasing cloud optical depth (an approach to an idealistic singularity of a straight vertical line). We determined that this effect was largely due to non-linearities in the spectra of the extinction efficiency and the 550 nm referencing of the cloud optical depth ($\tau$).

The optical properties of the liquid and dust particles were generated with a Mie Code (MiePlot4621, written by Philip Laven[21]) using the refractive index of water (Hale and Querry, 1973) and dust[22] for monodisperse particles. The optical properties of the ice crystals were extracted from Ping Yang's database (Yang et al., 2013) and correspond to a modified-gamma distribution with effective variance of 0.1 (Petty and Huang, 2011) of severely roughened column aggregates (Yang et al., 2013). This is the same distribution that is assumed in the Collection 6 MODIS cloud product (Holz et al., 2016).

Weak DODs associated with high-altitude Asian dust, the $BTD_{11-12}$ to DOD sensitivity ($dBTD_{11-12}$ / $dDOD$) would be best represented by a slope near DOD = 0 ($\tau = 0$ on the graphs) for the case of the near 1.5 µm peak radius of springtime Asian dust (see, for example, the right panel of Figure 16 of Burton et al, 2012 and Figure 3 of AeF for the springtime Asian dust particle size distribution). This yields a value (from the detailed numerical results employed in generating these graphs) of -0.30 °K per unit change in DOD.

The brightness temperatures correspond to the EOS-1 TERRA MODIS spectral response functions for bands 31 (max. at 11.0 µm) and 32 (max. at 11.9 µm), downloaded from the Satellite Application Facility for Numerical Weather Prediction (NWP SAF) website[23].

A nominal noise figure for MODIS Band 31 (the 11 µm band) is 0.05 °K (the cloud-discrimination ATBD of Team et al., 2010). Given a roughly equivalent (incoherent) noise for band 32 (the 12 µm band) yields a $BTD_{11-12}$ noise value of $\sqrt{2} \times 0.05$ 435 = 0.07 °K

## B.2 Choice of refractive indices at 11 and 12 µm

The refractive indices of water droplets and ice crystals are, as per the previous section, relatively well constrained and known. The observed dust refractive indices in the literature are principally dependent on dust composition (see, for e.g., Volz, 1972; Koepke et al., 1997; Rothman et al., 2009 and Sadrian et al., 2023): this dependence impacts the behavior of the $BTD_{11-12}$ vs 440 $BT_{11}$ patterns. A unique choice of refractive index based on dust composition is not possible given the diversity of dust types that characterize Asian and local dust over the Arctic (coupled with the often incomplete information on their composition).

---

[21] http://www.philiplaven.com/mieplot.htm
[22] see Appendix B.2 for a discussion of our choice of dust refractive index.
[23] https://nwp-saf.eumetsat.int/site/software/rttov/download/coefficients/spectral-response-functions/

Figure B2 illustrates the infra-red refractive index and (simulated) surface emissivity ($\varepsilon$) spectrum of water and ice particles as well as two distinctly different complex refractive indices of dust for two frequently referenced citations (Koepke, 1997 with principal components of quartz and clay and Volz, 1973 with principal components of clay, illite, and kaolinite). Those

different refractive indices result in significantly different emissivity spectral slopes between 11 and 12 μm.

We chose the Volz (1973) refractive indices essentially because the 11 to 12 μm spectral slope of the derived $\varepsilon$ values were of the same sign as the $\varepsilon$ slopes presented in VCT. This choice was underpinned by different levels of empirical and simulated evidence: the $BTD_{11-12}$ vs $BT_{11}$ "convex downward" pattern (generally indicating negative $BTD_{11-12}$ values) for a normal ($dT / dz < 0$) lapse rate (Figure B1 and Table 1) is coherent with satellite-based Asian dust measurements (as per footnote

g of Table 1) as well as the negative lapse rate and attendant negative $BTD_{11-12}$ across the optically thick local-dust Lake Hazen plume that was discussed above. Another level of empirical evidence in the Arctic was the significantly negative $BTD_{11-12}$ values in the dust emission regions of the drainage basins on Eglinton and Bank's Island (see Figure S8 and its caption). We also found moderately negative $BTD_{11-12}$ values in areas around Lake Hazen that were clearly the result of dust deposition on snow and ice.

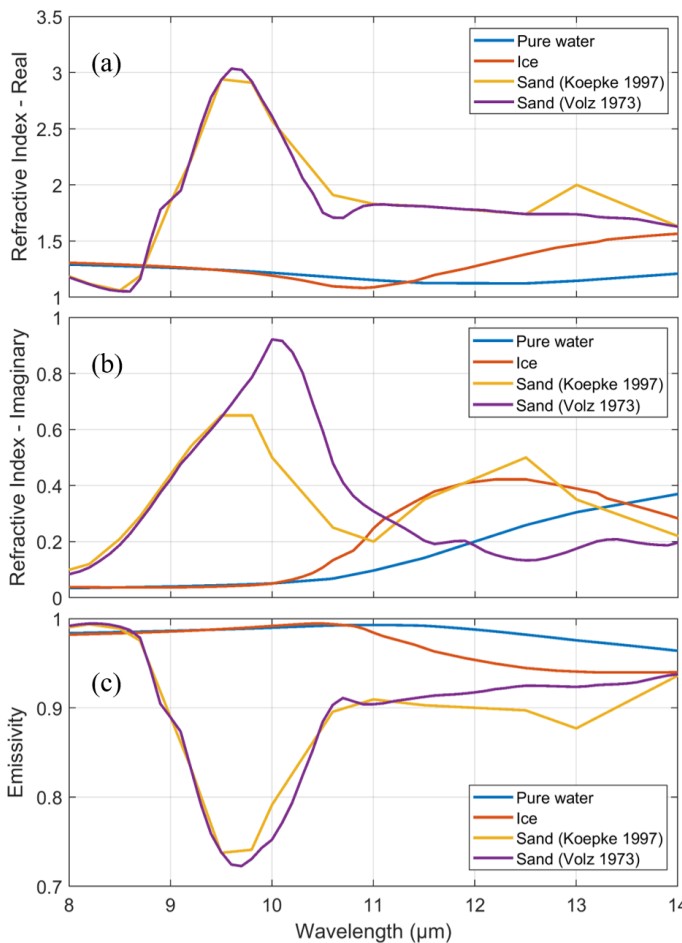


Figure B2 – Real (a) and imaginary (b) parts of the refractive index (*n* and *k*) and surface emissivity ($\varepsilon$) spectra (c) employed for the MODTRAN simulations of Appendix B1. The $\varepsilon$ spectra were approximately computed using the formulations of Masuda et al. (1988).

## Appendix C – Acronym and symbol glossary

| | |
|---|---|
| AERONET | World-wide NASA network of combined sunphotometer / sky-scanning radiometers manufactured by CIMEL Éléctronique. See AERONET website[27] for documentation and data downloads |
| AHSRL | Arctic High Spectral Resolution Lidar |
| AOD | The community uses "AOD" to represent anything from nominal aerosol optical depth which hasn't been cloud-screened to the conceptual (theoretical) interpretation of aerosol optical depth. In this paper we use it in the latter sense and apply adjectives as required. |
| AQUA | Polar orbiting NASA satellite whose payload includes the MODIS-Aqua multi-band imager. Aqua passes south to north over the equator in the afternoon (originally known as EOS PM-1) |
| ARM | Atmospheric Radiation Measurements |
| $\beta$ | Backscatter cross section ($m^{-1} sr^{-1}$) |
| BT, BTD | Brightness Temperature, Brightness Temperature Difference |
| CAA | Canadian Arctic Archipelago |
| CM | Coarse mode (generally referring to particles of super-micrometer radii) |
| COD | Cloud Optical Depth |
| DOD | Dust Optical Depth |
| FM | Fine mode (generally referring to particles of sub-micrometer radii) |
| HLD | High Latitude Dust |
| HYSPLIT | HYbrid Single-Particle Lagrangian Integrated Trajectory |
| INP | Ice Nucleation Particle |
| KAZRGE | Ka ARM Zenith Radar (KAZR)[28] GEneral mode. Zenith pointing Doppler radar operating at 35 GHz (8.6 mm) |
| LWP | Liquid Water Path |
| MISR | Multi-angle Imaging SpectroRadiometer |
| MODIS | Moderate Resolution Imaging Spectroradiometer |
| NSHSRL | North Slope High Spectral Resolution Lidar |

---

[27] https://aeronet.gsfc.nasa.gov/
[28] https://adc.arm.gov//metadata/html/nsakazrgeC1.b1.html

| Terra | Terra passes from north to south across the equator in the morning |
| TIR | Thermal InfraRed |

## 7 Author contribution

**Norm T. O'Neill:** Writing – original draft preparation – review & editing, Conceptualization, Methodology, Investigation, Formal analysis, Visualization, Validation, Project administration, Data curation, Funding acquisition, Resources. **Keyvan Ranjbar:** Writing – review & editing, Conceptualization, Investigation, Software, Formal analysis, Visualization, Validation. **Liviu Ivănescu:** Writing – review & editing, Conceptualization, Investigation, Software, Formal analysis, Visualization, Validation. **Yann Blanchard:** Writing – review & editing, Conceptualization, Investigation, Software, Formal analysis, Visualization, Validation. **Seyed Ali Sayedain:** Writing – review & editing, Conceptualization, Validation. **Yasmin AboEl-Fetouh:** Writing – review & editing, Conceptualization.

## 8 Competing interests

The authors declare that they have no conflict of interest.

## 9 Acknowledgements

This work was supported by the ESS-DA program of the Canadian Space Agency (CSA). We also gratefully acknowledge the longstanding research support provided by Natural Sciences and Engineering Research Council of Canada (NSERC). With respect to the Barrow (Utqiagvik) North Slope Alaska (NSA) analysis: data were obtained from the Atmospheric Radiation Measurement (ARM) user facility, a U.S. Department of Energy (DOE) office of science user facility managed by the Biological and Environment Research program. Valuable in-kind support was provided respectively by the AEROCAN network of Environment and Climate Change Canada (ECCC) and the NASA AERONET network. We also acknowledge the efforts of Antonis Gkikas of the National Observatory of Athens (NOA) Institute for Astronomy, Astrophysics, Space Applications and Remote Sensing (IAASARS ) in providing detailed MIDAS DOD retrievals and uncertainties over the Arctic.

## 10 Financial support

This research has been supported by the Canadian Space Agency (grant nos. 16SUASACIA and 21SUASACOA), and the Natural Sciences and Engineering Research Council of Canada (grant nos. RGPIN-2017-05531 and RGPIN-2023-04943).

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
