# Peer review of "Remote Sensing detectability of airborne Arctic dust"

_EGUsphere, 2024_

## Author Comment (AC1)

*General comment to the reviewer: the answers to your specific comments are immediately below. We would also point out that we made other changes that we thought would help to clarify the text (as well as the correction of few technical errors and/or typos that we missed in our proofreading before the original submission)*

Review of: "Remote Sensing detectability of airborne Arctic dust" by Norman O'Neill et al.

This paper basically describes the effect that based on brightness temperature dust in the Arctic may have been overestimated and misclassified in the past. It is basically a response to the Bowen & Vicent (2021) nature paper and in this work O'Neill et al. argue that water (clouds) may lead to an systematic overestimation of dust in the Arctic. This is of course a relevant finding. However, I am not very happy with the presentation. Due to the lack of a dedicated result section and a long supplement, which is not ordered I found the paper a bit difficult to read. I recommend to re-structure the paper for better readability.

> *We tried our best to respond to the "lack of a dedicated result section and a long supplement" comment. But before getting into details of how we responded to that comment we would remind the referee that our article is a general critique of overestimating DOD in the Arctic (with, as the reviewer indicated, a focus on one particular example). The type of results we use in our examples[1] can largely be found in the literature.*
>
> *We admit however that the $BTD_{11-12}$ vs $BT_{11}$ families of curves in the supplementary material should have been in an appendix of the paper (as per ACP protocols): we accordingly moved former Figure A8 and its discussion to a new sub-appendix of Appendix B (please see the new TOC at the end of this document). This provides the necessary detailed context for Table 1. What remains in the supplementary material are event details in support of various discussions in the main text (and notably animations that we think are important for the discussion of the Mar. 22, 2015 Barrow event).*

At some points I ask to provide more information, such that the reader can judge all statements, see my detailed remarks below.

- Line 42: (and later line 74) what do you mean by AOD 0.016 × 1.5 ±1 and ~ 0.0023 × 1.2 ±1 ? Is the exponent the Angstroem? If so: what does the +/- sign means? To what wavelength you are referring to? How did you derive these values?
  > *The AOD statistics employed in AboEl-Fetouh et al. (2020) are the monthly binned geometric mean and geometric standard deviation ("$\tau_{c,g}$" and "$\mu_c$") as applied to their coarse mode AOD databases). The geometric statistics at Resolute Bay[2] during the springtime (April & May) Asian-dust event are represented by $\tau_{c,g} \times \mu_c^{\pm 1} = 0.016 \times 1.5^{\pm 1}$. Similarly, the $\tau_c$ range during the June, July, and August summertime is $0.0023 \times 1.2^{\pm 1}$. The basics of geometric statistics as applied to CM AODs (as well as FM AODs) are outlined in the AeF citation (see the "Statistical Approach" of Section 3.4 in that paper).*
- Line 47: reference to S8: Tell here right away that MORTRAN has been used. Does this refer to an AOD of 1? Is the aerosol distribution log-normal (with which parameters?)
* * *
[1] *Including the case of negative $BTD_{11-12}$ for a water cloud in the inversion layer (which can be found in Key, 2002)*

[2] *in perfect analogy with the ubiquitous arithmetic mean / arithmetic standard deviation expression of $\langle \tau_c \rangle \pm \sigma(\tau_c)$*

*This is now a reference to the discussion of Figure B1 in Appendix B.1 (where MODTRAN is brought up in the 1st sentence of that appendix)*

- If you give AOD values without mentioning the wavelength, do you refer to 500nm?
  *The reference wavelength of 550 nm was stated explicitly in the footnote on page 2. For good measure we added the 550 nm explicitly in the sentence that contains the footnote in question*

- Line 73: provide units for "Arctic dust mass"

  *"Arctic dust mass" was changed to "Arctic dust mass" with units of mg-m$^{-2}$*

- Line 89: Ranjbar paper is it 2022 or 2021?
  *Citations to both Ranjbar papers (2021 and 2022) are correct (we double checked)*

- Given the general title of section 2, this paragraph is quite short. You may also relate to ground-based observations, which are in line with your argumentation, e.g. https://acp.copernicus.org/articles/17/8101/2017/

  *The reviewer's suggested paper is clearly a comprehensive analysis but it is rather removed from the issue of remote sensing detectability of columnar aerosol properties (short of performing some rather tedious optical / radiative transfer computations on particle size distribution profiles from the ground-based measurements (extrapolated to the vertical profiles) … which would be something less than the direct empirical optical proof that we sought in our paper). In response to the reviewer's "quite short" critique, we replaced our single sentence of (admittedly oversimplified) contextual text by a more substantive supporting narrative (citing a recent paper that we participated in on FM and CM AOD pan-Arctic reanalysis simulations). The first paragraph is now about climatological scale detection of Arctic aerosols while the 2nd paragraph is about the event level mis-interpretation by VCT*

- line 152: "Nearly all of our negative BTD …" This sentence is important and you should be more specific on your data.

  *This data has now been supplied in the form of supplementary material excel files*

- line 159: MODRAN simulations: you may list in the appendix the important values, assumptions etc. which you have used. Otherwise change the title to … in the IR from satellite.z

  *These details were now given in Appendix B.1*

- Table 1: I wonder, whether the relations between temperature lapse, emissivity slope and BTD pattern are always so easy and unique (if so: why?) what if there are several layers of dust and cloud in different altitudes? You must not fully answer this, but an idea of the assumptions and limitations behind the results of table 1 are important.

  *The technical details characterizing those "assumptions and limitations" are given in the text describing Figure B1 and the caption of Figure B1 in the new Appendix B.1.*

  *An explanation at the conceptual level (supported by the diagram below) begins with $BTD_{11-12}$ vs $BT_{11}$ curves being pinned to singular points ($\tau = 0$ and $\tau = \infty$). From a line joining those two*

points[3] *the curves spread concave up (BTD$_{11-12}$ > 0) or concave down (BTD$_{11-12}$ < 0): the BTD$_{11-12}$ sign depends on whether BT$_{11}$ or BT$_{12}$ from the ground is more penetrating into the cloud/plume[4] and whether dT/dz across the cloud/plume is negative or positive. No, we did not consider the case of dust and clouds at several altitudes but we do believe that the general behavior of such scenarios can be inferred from the single layer cases.*

[Figure]

- Line 251: Lhù'ààn Mân is that correct?

  *Yes. Lhù'ààn Mân' is the Southern Tutchone name for Kluane Lake. Southern Tutchone is one of seven Athapaskan languages in the Yukon and is spoken by Kluane First Nation people.*

- Table A2 is not completely clear to me. Can you please confirm or clarify: DOD_m in 3rd column is what you assume to calculate DOD(tau). The last column is what is needed to bring your results in agreement to KA. You are using eq A4 (not 4). If so: the DOD in last column is extremely variable (factor 16 between Resolute Bay and Eureka), I wonder how to interpret this. What are your critical assumptions here?

  *Our answer here (in order to minimize confusion for the reviewer) is based on the submitted version of the manuscript (before the re-arrangement of sections as described below).*

  *The last column of Table A2 is eliminated in the revised paper (it amounts to an unnecessary distraction that we regret having included[5]). Yes, $DOD_m$ is what we assumed to calculate $DOD$ ($\tau$). Everything we need to know is really in (old) Table A1[6]. Table A2 (without the irrelevant distraction of the last column) is just a retake of Table A1 with the added element of exploring the consequences of KA's supplementary material finding that their simulations have a negative bias ($\Delta DOD$) relative to CALIOP simulations of local dust. So the column $DOD - \Delta DOD$ addresses the issue of what happens if we apply a CALIOP-inspired correction.*
* * *
[3] *whose positions change, respectively with every surface and cloud/plume*

[4] *which in turn depends on the "point- volume" emissivity of particles in those two bands for a given type of cloud/plume (ice, water or dust). The relative inter-band, "point- volume" emissivity will be related to the surface emissivity (given for example in VCT's Figure 1)*

[5] *It was not an attempt "to bring your [our] results in agreement to KA": it was about an ultimately confusing distraction whose unnecessary goal was to explain the differences between the local dust and Asian dust values of $DOD_m$*

[6] *the new Table A1 is simplified by not being encumbered with repetitive $DOD_m$ information*

*In our revised paper we point out that, in actual fact, CALIOP "truth" in this case is simply unvalidated (in the end, we make an overarching statement that the amplitude of simulation differences (relative to the AeF climatology) are < 0.002*

- Line 336: cancel "a" between "yields" and "an"
  *The "a" was removed (note that that typo was located in line 356 rather than 336).*

- The value of appendix A6 and espec. Table A3 is not clear to me. I would be good to have a statement of the meaning in a broader context.

  *Indeed Appendices A5 and A6 contained technical information[7] that was unnecessary (to the point of creating unnecessarily confusing) and a distraction from the key narrative of simply providing DODs that characterized local dust (from Kawai et al.'s mass abundances) and Asian dust (for the April, 2001 event). These unnecessary $DOD_m$ arguments were removed from those two sections (and from Table A2) while retaining the adjusted Table A2 and Table A3 (and providing a broad context for both cases[8]).*

  *We also rearranged Appendix A in general to render its opto-physical development more "bottom up" (with more clarifying titles to accommodate this re-rearrangement as well as the transformation of old Appendices A.5 and A.6 into, respectively, new Appendices A.3 and A.4: a Word sample of the new appendix TOC is below). Appendix A.3 includes a new graph (Figure A1) which more explicitly (and clearly, we would argue) compares the AeF CM AODs with the KA DODs for the 4 AeF AERONET sites in the CAA (Canadian Arctic Archipelago) . There are still three tables labelled A1, A2 and A3: they have been revised to eliminate everything related to our misguided attempt to compare $DOD_m$ values of local and Asian dust.*

After these revisions I support a publication.
* * *
[7] *Most notably, the comparison of $DOD_m$ derived using historical Asian dust information with the $DOD_m$ value derived from the local dust mass abundances of Kawai et al. (2023)*

[8] *In the case of Table A3 we added the contextual sentence "This table supports the discussion surrounding the well documented Asian dust event of 2001 and the dust deposition consequences in the Mount Logan region."*

---

## Author Comment (AC2)

*General comment to the reviewer: the answers to your specific comments are immediately below. We would also point out that we made other changes that we thought would help to clarify the text (as well as the correction of few technical errors and/or typos that we missed in our proofreading before the original submission)*

Review of: "Remote Sensing detectability of airborne Arctic dust" by Norman O'Neill et al.

This paper basically describes the effect that based on brightness temperature dust in the Arctic may have been overestimated and misclassified in the past. It is basically a response to the Bowen & Vincent (2021) nature paper and in this work O'Neill et al. argue that water (clouds) may lead to an systematic overestimation of dust in the Arctic. This is of course a relevant finding. However, I am not very happy with the presentation. Due to the lack of a dedicated result section and a long supplement, which is not ordered I found the paper a bit difficult to read. I recommend to re-structure the paper for better readability.

*We tried our best to respond to the "lack of a dedicated result section and a long supplement" comment. But before getting into details of how we responded to that comment we would remind the referee that our article is a general critique of overestimating DOD in the Arctic (with, as the reviewer indicated, a focus on one particular example). The type of results we use in our examples[1] can largely be found in the literature that we cite or the illustrations that we give in the supplementary material.*

*We admit however that the $BTD_{11-12}$ vs $BT_{11}$ families of curves in the supplementary material should have been in an appendix of the paper (as per ACP protocols): we accordingly moved the $BTD_{11-12}$ vs $BT_{11}$ graphics of former Figure S12 and its discussion from the supplementary material to a new sub-appendix of Appendix B (please see the new TOC at the end of this document). This provides the necessary detailed context for Table 1. What remains in the supplementary material are event details in support of various discussions in the main text (and notably animations that we think are important for the discussion of the Mar. 22, 2015 Barrow event).*

At some points I ask to provide more information, such that the reader can judge all statements, see my detailed remarks below.

- Line 42: (and later line 74) what do you mean by AOD 0.016 × 1.5 ±1 and ~ 0.0023 × 1.2 ±1 ? Is the exponent the Angstroem? If so: what does the +/- sign means? To what wavelength you are referring to? How did you derive these values?
  *The AOD statistics employed in AboEl-Fetouh et al. (2020) are the monthly binned geometric mean and geometric standard deviation ("$\tau_{c,g}$" and "$\mu_c$") as applied to their coarse mode AOD databases). The geometric statistics at Resolute Bay[2] during the springtime (April & May) Asian-dust event are represented by $\tau_{c,g} \times \mu_c^{\pm 1} = 0.016 \times 1.5^{\pm 1}$. Similarly, the $\tau_c$ range during the June, July, and August summertime is $0.0023 \times 1.2^{\pm 1}$. The basics of geometric statistics as applied to CM AODs (as well as FM AODs) are outlined in the AeF citation (see the "Statistical Approach" of Section 3.4 in that paper).*
- *Line 47: reference to S8: Tell here right away that MORTRAN has been used. Does this refer to an AOD of 1? Is the aerosol distribution log-normal (with which parameters?)*
* * *
[1] *Including the case of negative $BTD_{11-12}$ for a water cloud in the inversion layer (which can be found in Key, 2002)*
[2] *in perfect analogy with the ubiquitous arithmetic mean / arithmetic standard deviation expression of $\langle \tau_c \rangle \pm \sigma(\tau_c)$*

*This is now a reference to the discussion of Figure B1 in Appendix B.1 (where MODTRAN is brought up in the 1st sentence of that appendix)*

- If you give AOD values without mentioning the wavelength, do you refer to 500nm?
  *The reference wavelength of 550 nm was stated explicitly in the footnote on the original and the new page 2. For good measure we added the 550 nm explicitly in the sentence that contains the footnote in question*

- Line 73: provide units for "Arctic dust mass"

  *"Arctic dust mass" was changed to "Arctic dust mass" with units of mg-m$^{-2}$*

- Line 89: Ranjbar paper is it 2022 or 2021?
  *Citations to both Ranjbar papers (2021 and 2022) are correct (we double checked)*

- Given the general title of section 2, this paragraph is quite short. You may also relate to ground-based observations, which are in line with your argumentation, e.g. https://acp.copernicus.org/articles/17/8101/2017/

  *The reviewer's suggested paper is clearly a comprehensive analysis but it is rather removed from the issue of remote sensing detectability of columnar aerosol properties (short of performing some rather tedious optical / radiative transfer computations on particle size distribution profiles from the ground-based measurements (extrapolated to the vertical profiles) … which would be something less than the direct empirical optical proof that we sought in our paper). In response to the reviewer's "quite short" critique, we replaced our single sentence of (admittedly oversimplified) contextual text by a more substantive supporting narrative (citing a recent paper that we participated in on FM and CM AOD pan-Arctic reanalysis simulations). The first paragraph is now about climatological scale detection of Arctic aerosols while the 2nd paragraph is about the event level mis-interpretation by VCT*

- line 152: "Nearly all of our negative BTD …" This sentence is important and you should be more specific on your data.

  *This data has now been supplied in the form of supplementary material excel files*

- line 159: MODRAN simulations: you may list in the appendix the important values, assumptions etc. which you have used.

  *These details were now given in Appendix B.1*

- Table 1: I wonder, whether the relations between temperature lapse, emissivity slope and BTD pattern are always so easy and unique (if so: why?) what if there are several layers of dust and cloud in different altitudes? You must not fully answer this, but an idea of the assumptions and limitations behind the results of table 1 are important.

*The technical details characterizing those "assumptions and limitations" are given in the text describing Figure B1 and the caption of Figure B1 in the new Appendix B.1.*

*An explanation at the conceptual level (supported by the diagram below) begins with BTD$_{11-12}$ vs BT$_{11}$ curves being pinned to singular points ($\tau = 0$ and $\tau = \infty$). From a line joining those two points[3] the curves spread*
* * *
[3] *whose positions change, respectively with every surface and cloud/plume*

*concave up (BTD$_{11-12}$ > 0) or concave down (BTD$_{11-12}$ < 0): the BTD$_{11-12}$ sign depends on whether BT$_{11}$ or BT$_{12}$ from the ground is more penetrating into the cloud/plume[4] and whether dT/dz across the cloud/plume is negative or positive. No, we did not consider the case of dust and clouds at several altitudes but we do believe that the general behavior of such scenarios can be inferred from the single layer cases.*

[Figure]

- Line 251: Lhù'ààn Mân is that correct?

  *Yes. Lhù'ààn Mân' is the Southern Tutchone name for Kluane Lake. Southern Tutchone is one of seven Athapaskan languages in the Yukon and is spoken by Kluane First Nation people.*

- Table A2 is not completely clear to me. Can you please confirm or clarify: DOD_m in 3rd column is what you assume to calculate DOD(tau). The last column is what is needed to bring your results in agreement to KA. You are using eq A4 (not 4). If so: the DOD in last column is extremely variable (factor 16 between Resolute Bay and Eureka), I wonder how to interpret this. What are your critical assumptions here?

  *Our answer here (in order to minimize confusion for the reviewer) is based on the submitted version of the manuscript (before the re-arrangement of sections as described below).*

  *The last column of Table A2 is eliminated in the revised paper (it amounts to an unnecessary distraction that we regret having included[5]). Yes, $DOD_m$ is what we assumed to calculate $DOD$ ($\tau$). Everything we need to know is really in (old) Table A1[6]. Table A2 (without the irrelevant distraction of the last column) is just a retake of Table A1 with the added element of exploring the consequences of KA's supplementary material finding that their simulations*
* * *
[4] *which in turn depends on the "point- volume" emissivity of particles in those two bands for a given type of cloud/plume (ice, water or dust). The relative inter-band, "point- volume" emissivity will be related to the surface emissivity (given for example in VCT's Figure 1)*

[5] *It was not an attempt "to bring your [our] results in agreement to KA": it was about an ultimately confusing distraction whose unnecessary goal was to explain the differences between the local dust and Asian dust values of $DOD_m$*

[6] *the new Table A1 is simplified by not being encumbered with repetitive $DOD_m$ information*

*have a negative bias ($\Delta DOD$) relative to CALIOP simulations of local dust. So the column $DOD - \Delta DOD$ addresses the issue of what happens if we apply a CALIOP-inspired correction. In our revised paper we point out that, in actual fact, CALIOP "truth" in this case is simply unvalidated (in the end, we make an overarching statement that the amplitude of simulation differences (relative to the AeF climatology) are < 0.002*

- Line 336: cancel "a" between "yields" and "an"

    *The "a" was removed (note that that typo was located in line 356 rather than 336).*

- The value of appendix A6 and espec. Table A3 is not clear to me. I would be good to have a statement of the meaning in a broader context.

    *Indeed Appendices A5 and A6 contained technical information[7] that was unnecessary (to the point of creating unnecessary confusion) and a distraction from the key narrative of simply providing DODs that characterized local dust (from Kawai et al.'s mass abundances) and Asian dust (for the April, 2001 event). These unnecessary $DOD_m$ arguments were removed from those two sections (and from Table A2) while retaining the simplified Table A2 and Table A3 (and providing a broad context for both cases[8]).*

    *We also rearranged Appendix A in general to render its opto-physical development more "bottom up" (with more clarifying titles to accommodate this re-rearrangement as well as the transformation of old Appendices A.5 and A.6 into, respectively, new Appendices A.3 and A.4: see the TOC below). Appendix A.3 includes a new graph (Figure A1) which more explicitly (and clearly, we would argue) compares the AeF CM AODs with the KA DODs for the 4 AeF AERONET sites in the CAA (Canadian Arctic Archipelago) . There are still three tables labelled A1, A2 and A3: they have been revised to eliminate everything related to our misguided attempt to compare $DOD_m$ values of local and Asian dust.*

- other things that we did to render the paper more understandable / readable .
    - *added a map of the CAA in the introduction*
    - *Provided a much more comprehensive explanation (Appendix B.2) about how we arrived at the refractive indices that were employed in the simulations of $BTD_{11-12}$ values*

- We provide below the new TOC of the Appendices so that the referee can better appreciate their re-arrangement (the sections of the main text did not change)

    After these revisions I support a publication.
* * *
[7] *Most notably, the comparison of $DOD_m$ derived using historical Asian dust information with the $DOD_m$ value derived from the local dust mass abundances of Kawai et al. (2023)*

[8] *In the case of Table A3 we added the contextual sentence "This table supports the discussion surrounding the well documented Asian dust event of 2001 and the dust deposition consequences in the Mount Logan region."*

---

## Author Comment (AC3)

The paper discusses the remote sensing of dust aerosols over the Arctic and the question of the possible misinterpretation of dust identification when using the brightness-temperature differences at 11 and 12 µm as parameter. In particular the possible bias induced by clouds in is investigated. The paper provides an interesting discussion against recent literature and provides illustration based on specific cases. The paper topic is well suited for ACP and surely of relevance for the dust and remote sensing community. However, the presentation quality should be improved before publication. As general comment, in fact, the paper is quite hard to read as the presentation and the discussion is based on many references to other papers, including mention to literature figures, and reference to Appendix and Supplementary material of the paper itself. Several footnotes are also present in the text and could be avoided. Despite it is appreciable to have a concise manuscript, the many references to literature and additional material in the paper make the reading often difficult. The reviewer suggests to revise the presentation to make it more self-consistent.

***General comment to the reviewer:*** *the answers to your specific comments are immediately below. We would also point out that we made other changes that we thought would help to clarify the text (as well as the correction of few technical errors and/or typos that we missed in our proofreading before the original submission)*

 *We made a concerted effort to respond to the general notion of our paper "making reading often difficult". This included the elimination of excessively detailed text that wasn't essential to the narrative of the paper:*

- *The paragraph in the main text that dealt with the "correction" of KA's (Kawai et al., 2023) simulations (involving the comparison with CALIOP estimates of local dust DOD). These are details that go beyond the (not very demanding) reasons for including a discussion of local dust simulation in the paper*
- *everything related to local vs Asian dust $DOD_m$ comparisons*
- *the appendix table with its admittedly outdated estimates of dust refractive index (i.e. the table that was entitled "Survey of dust refractive indices (11 and 12 µm)"). We replaced that table by simple refractive index + derived emissivity spectra (Figure B2 in Appendix B.2) that explicitly show and contextualize the refractive indices employed for all of our MODTRAN simulations of $BTD_{11-12}$*

*We also added a graph in Appendix A showing a comparison of the AERONET CM AOD (for the AERONET sites in or near the CAA) with DODs from the KA simulations: we believe that this lends support to the variety of arguments we make for the weakness of DODs in the Arctic (the main text was also clarified to underscore that point)*

*And we added other clarifying material:*

- *the addition of North American wide map in the main text as suggested by the referee.*
- *what we believe is a significantly clearer discussion related to the main text figure showing the radar profiles and the $BTD_{11-12}$ and $BT_{11}$ temporal series of Mar. 22, 2015 (that figure is now Figure 2 after the addition of the map as Figure 1)*

*We chose not to remove citations or footnotes: the referee will surely be open to the argument that as cumbersome as citations can sometimes be, they are endemic to a comprehensive scientific text. With*

*respect to footnotes, we believe that they actually make the text more readable and easier to understand: the reader is free to ignore footnotes in order to understand the higher level narrative of a given text (and free to consult the footnotes if he or she feels the need to dig deeper into the technical details)*

Other specific comments:

Section 2 is quite short and not fully clear in particular since, as the introductory part, it relies on the reference to literature and supplementary material

> *We replaced our single sentence of (admittedly oversimplified) contextual text by a more substantive supporting narrative. The first paragraph is now about climatological scale detection of Arctic aerosols while the 2$^{nd}$ paragraph is about our claims concerning the event level mis-interpretation by VCT*

Introduction and following sections: many literature measurements from diverse sites in the Arctic are discussed. It would be good to have the localisation of these sites either in the form of latitude and longitude (in Table or main text; these are mentioned for some sites in the Appendix section only) or as a map. A map could be useful to provide some contextualisation of the discussion for a non-Arctic specialized reader.

> As stated above, w*e added a map (new Figure 1) : we think that the reviewer is correct and that it does indeed give contextual colour (breathing space) to our short but technically dense main text*

Appendix A, line 293: the value of $reff$ of 2.7 µm is referring to a transported or locally emitted dust? As the dust diameter changes over transport time due to gravitational settling, is this assumed Reff value representative of source or long range transported dust? Please clarify in the text.

> *Text line was clarified (the Kok distribution, of $r_{eff}$ = 2.7 µm, refers to locally emitted dust). Kawai-derived  estimates of DODs that would, at a distance from the emissions, be smaller due to smaller-sized dust particles, will not impact our order-of-magnitude claim that the Kawai-based DODs are substantially smaller than the AeF estimates.*

Footnote number 7 and Appendix B1: the OPAC database is quite outdated to represent dust infrared refractive index and the survey in Table B1 is missing several key works in the literature that investigated the infrared refractive index of dust aerosols. For this reason, I would either change the title of this section to clarify that this is not an exhaustive survey, or to extend the survey and take the variability of the refractive index of dust into account.

> *As stated above, we deleted that largely irrelevant refractive index survey table in favour of graphical refractive index and emission spectra which showed and contextualized the explicit refractive indices employed in our MODTRAN simulations of BTD$_{11-12}$.*

Final note to the referee

> *In response to the 2$^{nd}$ referee, we provided a more comprehensive explanation of the parameterizations employed to generate the BTD$_{11-12}$ vs BT$_{11}$ patterns for ice, water and dust clouds (and moved that figure + its discussion from the supplementary material to a new Appendix B.1 [supported by the new Appendix B.2 where the choice of refractive index is justified])*

*We also rearranged Appendix A in general to render its opto-physical development more "bottom up" (with more clarifying titles to accommodate this re-rearrangement as well as the transformation of old Appendices A.5 and A.6 into, respectively, new Appendices A.3 and A.4: a Word sample of the new Appendices TOC is below). Appendix A.3 includes a new graph (Figure A1) which more explicitly (and clearly, we would argue) compares the AeF CM AODs with the KA DODs for the four AeF AERONET sites in the CAA (Canadian Arctic Archipelago). There are still three tables labelled A1, A2 and A3: they have been revised to eliminate everything related to our misguided (unnecessarily complicating) attempt to compare $[DOD]_m$ values of local and Asian dust.*

*We provide below the new TOC of the Appendices so that the reviewer can better appreciate their re-arrangement (the TOC of the main text did not change)*